# The virtual microbiome: A computational framework to evaluate microbiome analyses

**Belén Serrano-Antón**[1,¤a,¤b], **Francisco Rodríguez-Ventura**[1], **Pere Colomer-Vidal**[1], **Riccardo Aiese Cigliano**[2]*, **Clemente F. Arias**[1,3]*, **Federica Bertocchini**[1]*

**1** CIB, Centro de Investigaciones Biológicas Margarita Salas (CSIC), Madrid, Spain, **2** Sequentia Biotech SL, Barcelona, Spain, **3** Grupo Interdisciplinar de Sistemas Complejos de Madrid (GISC), Madrid, Spain

¤a Current address: FlowReserve Labs SL, Santiago de Compostela, Spain
¤b Current address: Group of Nonlinear Physics, University of Santiago de Compostela, Santiago de Compostela, Spain
* raiesecigliano@sequentiabiotech.com (RAC); tifar@ucm.es (CFA); federica.bertocchini@csic.es (FB)

**Data Availability Statement:** All relevant data are within the paper and its Supporting information files.

## Abstract

Microbiomes have been the focus of a substantial research effort in the last decades. The composition of microbial populations is normally determined by comparing DNA sequences sampled from those populations with the sequences stored in genomic databases. Therefore, the amount of information available in databanks should be expected to constrain the accuracy of microbiome analyses. Albeit normally ignored in microbiome studies, this constraint could severely compromise the reliability of microbiome data. To test this hypothesis, we generated virtual bacterial populations that exhibit the ecological structure of real-world microbiomes. Confronting the analyses of virtual microbiomes with their original composition revealed critical issues in the current approach to characterizing microbiomes, issues that were empirically confirmed by analyzing the microbiome of *Galleria mellonella* larvae. To reduce the uncertainty of microbiome data, the effort in the field must be channeled towards significantly increasing the amount of available genomic information and optimizing the use of this information.

## Introduction

The characterization of the microorganisms colonizing a particular ambient is becoming a gateway to the analysis of the physiological niche that the environment represents, revealing its potential functions or eventual pathological conditions. Examples in this direction are represented by the deep interest in the human gut microbiome (the compendium of microorganisms colonizing the human gut), due to the growing concern in the relationships between the microbiome and the immune system, and henceforth in the potential development of disease [1–4]; or the increasing focus on the genomic analyses of water (ocean or river) and soil samples, in search for potentially useful cataloging of the environmental niches we live in [5].

Microbiome studies are receiving increasing attention for their possible implications in the field of bioremediation, embracing issues such as degradation of organic chemicals, conversion of toxic compounds (e.g. pesticides), production of biofuels, or else from various

**Funding:** FB and CFA gratefully acknowledge support by the Roechling foundation. BS was partially supported by MINECO grant MTM2017-85020-P. The funders had no role in study design, data collection and analysis, decision to publish, or preparation of the manuscript.

**Competing interests:** The authors have declared that no competing interests exist.

substrates [6–9]. The strong interest in communities of microorganisms crossed fields and extended to the studies of insect guts [10, 11]. A growing body of research focuses on insect microbiomes in the quest for solutions within the bioremediation field [12]. For example, termites and beetles have raised interest due to their capacity to degrade lignin, an ability potentially dependent in some cases on microorganisms colonizing certain portions of their intestine [13–17]. Recently, some coleopteran and lepidopteran species revealed the astonishing capability to degrade fossil fuel-derived plastics, like the sturdy polyethylene and polystyrene [18–21], opening up a new niche within the field of bioremediation by insects. Even if the molecular mechanisms responsible for this extraordinary capacity are still unknown, they are normally ascribed to the microorganisms colonizing the digestive tracts of those insects. This line of research has resulted in an ever-increasing list of microorganisms with the potential to biodegrade plastics, although with still un-concluding outcomes [20–26].

Studies in this field typically monitor the changes induced by alternative treatments in the relative abundance of the species present in the microbiome. For instance, feeding insects a diet of plastic is a standard procedure to identify the microorganisms that thrive in their gut as potential candidates for plastic metabolization [18, 20, 23, 26–28]. Analogously, the microbiome of patients suffering a given clinical condition is often compared to that of healthy control individuals, expecting the shifts in the relative abundance of microbial species to account for the observed effects or to provide effective diagnostic tools [29]. This functional approach to the study of microbiomes relies on several implicit assumptions, whose general validity is far from evident. First, a causal link is presumed between changing conditions and the differential selection (either positive or negative) of particular microbial species. In the case of plastic fed-insects, for instance, possible metabolic changes induced by a diet of plastic in the host insect are normally neglected, implying that the microorganisms colonizing the digestive tracts are considered responsible for any metabolic activity the animal embarks on.

Leaving aside its biological plausibility, this perspective takes for granted other assumptions that have to do with the methods used to study microbiomes. These methodological assumptions are the focus of this work. It is normally presupposed that currently available techniques are accurate enough to provide reliable quantitative measurements of changes in microbiome composition. Consequently, observed variations in the abundance of species (or other relevant taxa) are regarded as valid indicators of the activity of interest. A close inspection of the currently available empirical tools reveals unexpected and critical drawbacks in this approach.

The most used methodologies to study the microbiome of a chosen animal species are amplicon and whole-genome sequencing (WGS). The former is based on the identification of taxonomical markers such as the ribosomal gene 16S for bacteria. Although this relatively short gene ($\sim$1500 bp) is universal among bacteria and archaea [30], some studies have revealed that 16S rRNA gene does not show precise phylogenetic relationships within particular taxa [31, 32]. For this reason, in recent years there has been an increase in the number of analyses carried out with WGS, which is a more expensive and computationally intensive technique than amplicon.

Amplicon and WGS analyses rely on the comparison of DNA sequences obtained from a sample of the microbial population of interest with the sequences contained in genomic databanks. Therefore, they can only identify those sequences that are already present in databases. In addition, whole genome sequencing data can also be used to generate genome assemblies of the microbial populations, the so-called MAGs (Metagenome Assembled Genomes) which can be used to reconstruct the genome of new species. However, also in this case, databases of known species are used to try to classify the taxonomy of the MAGs. As a consequence, the accuracy of these methods should be expected to depend on the number of sequences contained in the databases used for the analysis.

Remarkably, comparative studies have shown that the results of the characterization of bacterial populations by amplicon and WGS do not necessarily overlap even if they use the same reference databanks. For example, applying both techniques on the same DNA from a human fecal sample resulted in a mere 29% overlapping at the phylum level [33], and revealed differences in the metagenomic results for a human salivary sample, both in the relative abundances of species and in the identified genera [34]. Even the same method (WGS) analyzed with different pipelines yielded results that differ by up to three orders of magnitude on the same data set [35].

In addition to the issues related to the methodology used to characterize microbiomes, the ecological structure of bacterial populations (e.g. diversity or richness of species) and the sampling strategy can also bias their characterization, adding further concerns about the accuracy of microbiome analyses [36, 37]. This scenario raises the following issues: To what extent and under what circumstances can we count on the current experimental approaches to understanding the complexity of symbiotic microorganisms living within the gut (or any other tissue) of an animal? Do the available techniques have intrinsic limitations that might influence the analysis of any microbiome, whatever the colonized environment? Or are these limitations restricted to particular habitats?

To get a deep insight into this puzzling landscape, we took an *in-silico* approach, creating virtual microbiome communities that simulate the bacterial populations that can be found in humans, insects, etc., or soil, water, and any other medium. These virtual communities can be examined by standard genomic techniques, giving the possibility to confront the results of the analysis with the original microbiome composition.

Within the virtual microbiome framework, it is possible to simulate the analysis of microbial populations with partially underrepresented or limited databanks, and hence to explore the impact of the lack of genomic information on the characterization of microbiomes. Using this approach, we have identified the incompleteness of databases as a key constraint to the accuracy of microbiome analyses. Our results suggest that the poor overlap between amplicon and WGS already encountered with human samples does not seem to be an exception. Furthermore, we show in this work that the simultaneous detection of a species by both techniques does not ensure its positive identification, since the overlap between amplicon and WGS can be dominated by false positives, i.e. by species that are not present in the original samples. This counterintuitive situation worsens notably when limited databases are used in the analyses to simulate the lack of genomic information. In this case, the detection of changes in the relative abundance of bacterial groups can be profoundly misleading. The variations in abundance detected by genomic analyses can be mostly spurious and unrelated to the actual changes taking place in the analyzed virtual microbiomes.

To empirically test the predictions derived from our model, we proceeded with the analysis of the bacteria colonizing different tissues of the larvae of *Galleria mellonella*, a lepidopteran recognized as capable of degrading polyethylene and polystyrene [19, 23–25, 38–41]. Despite their abundance, the microbiota of lepidopteran species has been little studied [42, 43], so the bacteria associated with this group have only a marginal representation in databanks to date. This makes *G. mellonella* larvae an ideal subject to examine the limitations of genomic analyses pinpointed by the virtual microbiome model.

To do that, we applied amplicon sequencing (16S) and WGS (using the same DNA samples) to the anterior part, the gut, and the silk glands of plastic-treated and control larvae. The results reflected the incongruences revealed by our model, with an astonishing lack of overlap between the outcomes of the two techniques at the genus level. Both techniques also showed striking discrepancies in the detection of changes in the insect microbiome.

For those animal species and environments that harbor microbial populations [44], microbiome analysis stands as a fundamental tool in the comprehension of their physiology and

ecology. However, the shortcomings inherent to the used tools and the paucity in the available database hinder and mislead the outcome, and, therefore, the interpretation of microbiome analyses. This work highlights the need for an increased effort in the collection of genomic information and its eventual availability in public databases. In the meantime, the results of experimental studies about the composition or the dynamics of microbiome populations should be interpreted with caution, especially in the case of insects, where the available genomic information is still scarce.

# Materials and methods

## Generation of virtual microbiomes

Virtual microbiomes were created with a fixed number of species (300). The process of generating abundance matrices was automated by means of an algorithm that takes as input the relevant parameters of the problem (i.e. the number of populations and species and the shape parameters of the distributions, labelled as *params*) and outputs an abundance matrix (labelled as *abundances*) that complies with the macroecological laws followed by real-world microbial populations (as defined in [36]). The full implementation of the algorithm can be found at the following link https://github.com/bserranoanton/Virtual-microbiome. In addition, information on all the parameters used for the generation of the abundance matrices of the virtual microbiomes can also be accessed. A simplified pseudocode of the algorithm is shown below. The resulting distributions were tested using the Kolmogorov-Smirnov test for goodness of fit.

**Algorithm 1** Generation of the ecological structure of the virtual microbiomes.

```
 1: nFamilies ← nFamilies ▷Number of families
 2: nGenus ← nGenus ▷Number of genus
 3: nSpecies ← nSpecies ▷Number of species
 4: nCommunities ← nCommunities ▷Number of communities
 5: muMAD ← mu ▷mu parameter value for abundance distribution
 6: sigmaMAD ← sigma ▷sigma parameter value for abundance distribution
 7: shapeAFD ← shape ▷shape parameter value for abundance distribution
 8: scaleAFD ← scale ▷scale parameter value for abundance distribution
 9: muGenus ← mu ▷mu parameter value for grouping species into genera
10: sigmaGenus ← sigma ▷sigma parameter value for grouping species
into genera
11: muFamilies ← mu ▷mu parameter value for grouping genera into
families
12: sigmaFamilies ← sigma ▷sigma parameter value for grouping genera
into families
13: procedure GENERATEMAD(mu, sigma, n)
14:   MADvector ← random.lognormal(mu, sigma, n)
15:   MADvector ← MADvector/sum(MADvector)
16:   return MADvector
17: procedure GENERATEAFD(shape, scale, n)
18:   AFDvector ← random.gamma(shape, scale, n)
19:   AFDvector ← AFDvector/sum(AFDvector)
20:   return AFDvector
21: procedure GENERATEABUNDANCEDISTRIBUTION(mu, sigma, shape, scale, nCommu-
nities, nSpecies)
22:   MADvector ← generateMAD(mu, sigma, nSpecies)
23:   abundances ← matrix(nCommunities, nSpecies)
24:   for species_i ← 1 to nSpecies do
25:     AFDvector ← generateAFD(shape, scale, nCommunities)
26:     abundances[:,species_i] ← MADvector[species_i] * AFDvector *
nCommunities
27:   return [MADvector, abundances]
```

```
28: procedure EcologyGeneration(nCommunities, nSpecies, mu, sigma, shape,
scale)
29:    [MADvector, abundances] ← generateAbundanceDistribution(muMAD,
sigmaMAD, shapeAFD, scaleAFD, nCommunities, nSpecies)
30:    r_value ← testTaylorsLaw(MADvector, abundances) ▷Continue if
correlation coefficient between mean and variance ≥0.95
31:    alphaDiv, betaDiv ← getMeanDiversities(abundances) ▷And clas-
sify scenario in high or low diversity
32:    genusAbundances ← groupSpeciesInGenera(muGenus, sigmaGenus,
nGenus, nSpecies) ▷Groups nSpecies in nGenus following a lognormal
distribution
33:    familyAbundances ← groupGeneraInFamilies(muFamilies, sigmaFami-
lies, nFamilies, nGenus) ▷Groups nGenus in nFamilies following a log-
normal distribution
```

## Genetic composition of virtual microbiomes

The species of virtual microbiomes were chosen from eight families: Clostridiaceae, Flavobacteriaceae, Enterococcaceae, Pseudomonadaceae, Acetobacteraceae, Lactobacillaceae, Enterobacteriaceae and Streptococcaceae. Species are grouped into genera and genera are grouped into families using lognormal distributions. This means that the number of genera in each family and the number of species in each genus follow lognormal distributions.

To generate the complete databank, 16S sequences from all prokaryotes and genome sequences were downloaded on February 2021 and April 2020 from NCBI Nucleotide, respectively (see S1 Data in S2 Appendix). To simulate the lack of information in genomic databases, we created incomplete databases by removing taxonomic information from this databank. We created three different databanks (see S1 Data in S2 Appendix):

1. Deleting half of the families (Incomplete database DB1).

2. Deleting half of the genera (Incomplete database DB2).

3. Deleting half of the species (Incomplete database DB3).

To simulate the lack of information in the databanks about the species present in the virtual microbiomes, we created three scenarios (S2–S4 Data in S2 Appendix):

1. 100% scenario: All the genomic information of the virtual microbiomes are present in the incomplete database.

2. 50% scenario: Only half of the genomic information of the virtual microbiomes are present in the incomplete databases.

3. 25% scenario: Only a quarter of the genomic information of the virtual microbiome are present in the database.

We remark that all the species in the virtual microbiomes are present in the complete database. For scenarios 2 and 3, we selected 50% and 75% of the taxonomical information from scenario 100% and replaced it with information that was not in the corresponding database, taking into account the restrictions exposed in 2.5. For the scenarios with 50% and 25% of the species present in the databases, there are several ways to remove families. In the case of removing families, the choice was made to remove those families that grouped an abundance of around 50% in the first case and 25% in the second (as close as possible within the restrictions exposed in 2.5). Therefore, the 50% scenario gives rise to two scenarios (A and B) and the 25% scenario to four (A, B, C and D). Full information on the taxonomic composition of each scenario can be found in S2–S4 Data of S2 Appendix.

## Bioinformatic analysis of virtual microbiomes

The number of reads corresponding to each species was proportional to its abundance in each virtual microbiome. We considered two scenarios regarding the total number of reads: high number of reads (100,000 for 16S and 60,000 for WGS) and low number of reads (50,000 for 16S and 20,000 for WGS). The generation of reads was done with the program Grinder [45] with the following parameters:

**Listing 1**. Grinder commands for reads generation.

```
#16S reads
grinder -reference_file microb_16S seq/$name.fasta
  -forward_reverse primer16_v2. fasta -total_reads
  $abundanceTotal -read_dist 300 -insert_dist 550 normal 55
  -unidirectional 1 -length bias 0 -mutation_dist poly4 -fq 1
    -qual_levels 30 10 -base_name $name -output_dir ./ reads16S
#WGS reads
grinder -reference_file microb_WGS_seq/$name.fasta -total_reads
    $abundanceTotal -read_dist 150 -insert_dist 400 normal 40
    -unidirectional 0 -length_bias 0 -mutation_dist poly4 -fq
  1 -qual_levels 30 10 -base_name $name -output_dir ./
  readsWGS
```

Reference files (*reference_file*) store all existing sequences for each species in the complete database. In this way, reads are generated for all the species avoiding any bias in the choice of a particular sequence. *total_reads* refers to the number of reads needed for each species. These reads were analysed by GAIA [46] using the complete and incomplete databases.

For amplicon, the length of reads (*read_dist*) was set at 300 base pairs (bp), with an overlap of 50 bp (*insert_dist* of 550 bp). For WGS the length of the reads was 150 bp and *insert_dist* of 400 bp. We used unidirectional reads (from one strand only) for amplicon and bidirectionally for WGS, as recommended by Grinder documentation (https://sourceforge.net/projects/biogrinder/).

In addition, we introduced sequencing errors in the reads (*mutation_dist*), under the form of mutations (substitutions, insertions and deletions) at positions that follow a 4*th* degree polynomial, simulating Illumina errors [47]. The quality of the reads (*qual_levels*) varies from good (30) to bad (10), for reads with insertions or substitutions. We tried to avoid length bias (i.e. greater contribution of reads from larger genomes) by setting *length_bias* to 0.

Primers used for the amplification of the V3V4 region in amplicon (parameter *forward_reverse*) were:

```
>V3V4F:34
CCTACGGGNGGCWGCAG
>V3V4R:34
GGACTACHVGGGTATCTAATCC
```

These primers have been shown to be efficient for 16S analysis [48]. However, these sequences were not able to amplify the desired region in a non-negligible percentage of the species. For this reason, a lower number of reads than expected was obtained in some *in silico* amplifications. To reach the desired number of reads, the remaining percentage of abundance was distributed according to the abundance distribution. Metagenomic analysis of the samples was done with GAIA [46], which allows to obtain a comprehensive and detailed overview at any taxonomic level of microbiomes in an accurate and easy way.

To confirm the results obtained with GAIA, a second set of virtual microbiomes was analyzed with Kraken version 2.1.1 [49]. The databases were downloaded from the repository created by the Kraken2 developers (https://benlangmead.github.io/aws-indexes/k2). Specifically, the 16S database was the Silva138 whereas the WGS database was the Standard (Date 9/26/2022). The Kraken2 classification analysis was performed with default parameters.

Owing to the low overlap between Kraken2 and GAIA databases, it was not possible to analyze the same virtual microbiomes with both tools. The virtual microbiomes analyzed with Kraken were created as described above for GAIA. For Kraken2, only the scenario of 100% species in the databases was considered. Kraken2 analyses were made at the species and genus level for WGS and only at genus for 16S, as this tool does not distinguish species. The virtual microbiomes analyzed with Kraken2 can be found in S9 Data of S2 Appendix.

## Analysis of the microbiome of *Galleria mellonella* larvae

*Galleria mellonella* larvae were maintained in an incubator at 28 C in the dark, and fed with beeswax from beehives.

For the experimental samples, *G. mellonella* larvae from the last larval stage (150–200mg) were kept in the presence of commercial plastic films only, during 4 to 6 days. For tissue extraction, live larvae were washed with 70% ethanol and rinsed with sterile water. Dissections were performed using sterilized microsurgery tools. DNA extraction was performed using the Quick-DNATM Tissue/Insect Miniprep Kit (Zymogen Research, Cat. No. D6016). Control larvae fed with beeswax from beehives were treated in the same way.

Amplicon sequencing was perfomed using the V3V4 primers described above to generate 16S Illumina libraries which were then sequenced with a MiSeq producing 300 bp paired-end reads. The same DNA was also used to generate TruSeq PCR-Free DNA libraries to perform whole metagenome sequencing (WGS) with an Illumina Novaseq 6000 producing 150 bp paired-end reads. Both the 16S amplicon sequencing data and the WGS data was processed with the following approach: first, the quality of the reads was assessed with the software FastQC (https://www.bioinformatics.babraham.ac.uk/projects/fastqc/), then low quality bases and adapters were removed with the software Trimmomatic (http://www.usadellab.org/cms/?page=trimmomatic); minimum quality 25 and minimum length 35 bp). Amplicon data was analyzed with the software GAIA as described above. WGS data was first mapped against the *G. mellonella* reference genome (ASM258982v1) using the software minimap2 (https://github.com/lh3/minimap2) to remove the contamination from the host. Unmapped reads were extracted with samtools (http://www.htslib.org/) and then processed with GAIA to perform the taxonomic classification. Differential abundance analyses were performed with edgeR (https://bioconductor.org/packages/release/bioc/html/edgeR.html) applying the TMM normalization to the abundance matrix of the OTUs. The results of the amplicon and WGS analysis of the microbiome of *G. mellonella* can be found in S5 and S6 Data of S2 Appendix respectively. S7 and S8 Data in S2 Appendix show the taxa whose abundance changes significantly in larvae fed with plastic in 16S and WGS analysis respectively. The results of the Kraken 2 analyses of the *G. mellonella* gut microbiome can be found in S10 Data of S2 Appendix.

## Results

### Virtual microbiome models

Virtual microbiome are models of bacterial populations conceived to evaluate the performance of genomic analyses. Each virtual microbiome consists of a list of species of bacteria and their respective abundances. Their ecological structure (i.e. the taxonomic relationships among species and their relative abundance) was designed to comply with the macro-ecological rules described for real-world microbial communities (see [36] and S1 Appendix).

To study how the incompleteness of genomic databanks affects microbiome analyses, we created restricted (incomplete) databases in which some families, genera, or species were removed from the complete database (i.e. the one containing all the prokaryotes genetic sequences available to date). This allowed us to simulate different scenarios in which some of

the species present in a microbiome have not been previously cataloged. In particular, we generated virtual microbiomes for which 25%, 50% and 100% of the species are represented in the incomplete databases.

Simulated amplicon sequencing and WGS reads were then generated starting from the virtual microbiomes using complete and incomplete databases and different numbers of reads (see Methods). Thus, for each virtual microbiome, we obtained several characterizations that could be compared with the original virtual microbiome composition and with one another (see Fig 1). First, we evaluated how the incompleteness of genomic databanks affects the accuracy of microbiome analyses using the following variables:

1. Unsampled species and genera: percentage of species and genera in the virtual microbiome whose abundance is below the threshold of detection imposed by the number of reads used in the analysis.

2. Unidentified species and genera: percentage of the species and genera that were sampled but did not match any of the sequences contained in the database used in the analysis.

3. False positives: percentage of the total of species and genera detected by the analysis that were not present in the original virtual microbiome.

4. True positives: percentage of the species and genera that were detected by the analysis.

5. Abundance of species and genera predicted by the metagenomic analysis.

Second, we studied the ability of microbiome analyses to detect changes in the abundance of species and genera in the virtual microbiomes. To that end, we compared virtual microbiomes containing the same set of species but differing in their relative abundance, a scenario that simulates the temporal changes in a microbiome caused, for instance, by alternative treatments on the host. Finally, we measured the overlap between 16S and WGS analyses (i.e. the species and genera simultaneously detected by both techniques) when performed on the same virtual microbiomes.

## Effect of the incompleteness of databases in the characterization of virtual microbiomes by microbiome analyses

The first step of microbiome analyses consists in sampling the microbial populations of interest. The limited size of samples imposes obvious constraints to the subsequent genomic

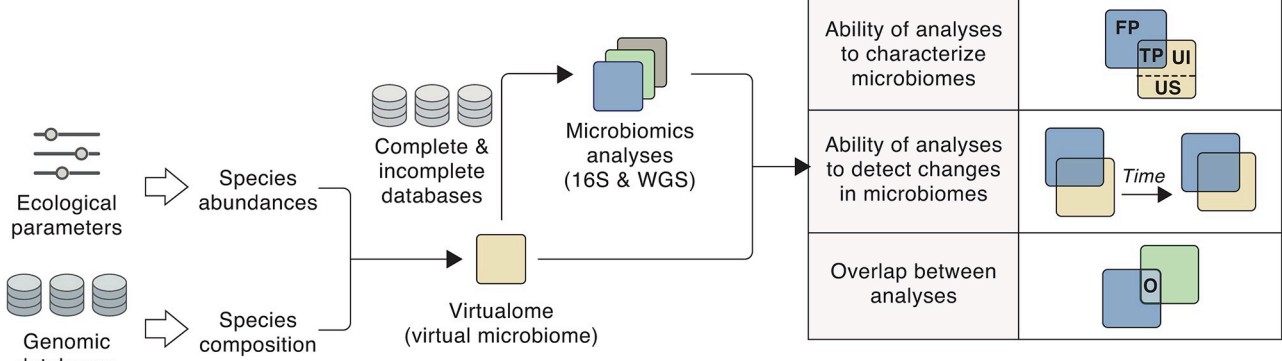

**Fig 1. Rationale of the virtual microbiome framework.** See explanations in the text. (FP: false positives; TP: true positives; UI: unidentified; US: unsampled; O: overlap.).

analyses of microbiomes: these analyses are performed only on the fraction of the microorganisms present in the microbiome that are actually sampled, which may hinder the identification of species with lower abundances [36].

To test the effect of sampling on the performance of microbiome analysis, we generated communities that differ in both species richness and $\alpha$-diversity. Smaller species richness and greater $\alpha$-diversities resulted in smaller fractions of unsampled species (Fig 2A left). In contrast, low-diversity communities are dominated by a few species or genera. Unevenness in the relative abundance implies that rare species are likely to be missed by the analysis. In these cases, increasing the number of reads only led to a moderate growth in the percentage of species detected (light grey lines, corresponding to this type of communities, do not fall below 80% in Fig 2A). This effect was more pronounced in more species-rich populations, in which greater sampling efforts did not necessarily result in a substantial reduction in the number of unsampled species, even for high values of $\alpha$-diversity (Fig 2A right). The ecological structure of microbial communities (in terms of richness of species and $\alpha$-diversity) imposes a major intrinsic limitation to the capacity of microbiome analyses, as shown by the loss of over 80% of the species in virtual microbiomes with low diversity (Fig 2B). In the subsequent evaluation of microbiome analyses, we used communities with low species richness (containing only 300 species) to minimize the fraction of unsampled species.

Unexpectedly, the incompleteness of genomic databanks led to a sharp increase in the detection of false positives (percentage of detected species or genera detected that are not present in the original virtual microbiome). Fig 2C shows that most of the analyses yielded high fractions of false positives (both at the species and genus levels) even when they were performed using the complete databank. This situation dramatically worsened for virtual microbiomes containing new species (corresponding to analyses with incomplete databases), in which false positives may represent more than half of the identified species and genera (Fig 2C). Remarkably, the incompleteness of the databanks affected the analyses of virtual microbiomes even when the 100% of their species were represented in the incomplete databases (the open dots do not lie in the diagonal in Fig 2C). The detection of false positives is determined by the presence in the reference databank of similar sequences to those sampled from the virtual microbiomes. Greater databanks may contain more sequences exhibiting similarities with virtual microbiome sequences, thus increasing the chances of misidentification. Conversely, if the number of potential candidates decreases, so can do the number of false positives.

We next analyzed the ability of genomic analysis to detect true positives (percentage of the species and genera of the virtual microbiome successfully identified in genomic analyses). This percentage was below 50% in all the virtual microbiomes at the species level (Fig 2D). As should be expected, the detection of true positives increased when larger fractions of the virtual microbiome species were included in the genomic databases used in the analysis. At the genus level, the detection of true positives was highly variable, ranging from around 10% to 100% in the studied virtual microbiomes.

Genomic analyses of virtual microbiomes provided very poor estimates of the abundance of species and genera. The differences between the actual abundance and that observed by the analyses spanned several orders of magnitude (Fig 2E). Remarkably, the results of this analysis were worse at the genus than at the species level, contrary to the characterization of unidentified, true positives, and false positives.

In view of their inability to accurately estimate the abundance of species and genera, it was not surprising that genomic analyses normally failed to detect changes in that abundance. We measured the success of the analyses to identify trends (i.e. increases or decreases in the abundance of species and genera), regardless of their magnitude. When incomplete databases were used in the analysis, most of the observed trends were spurious, i.e. they did not correspond to

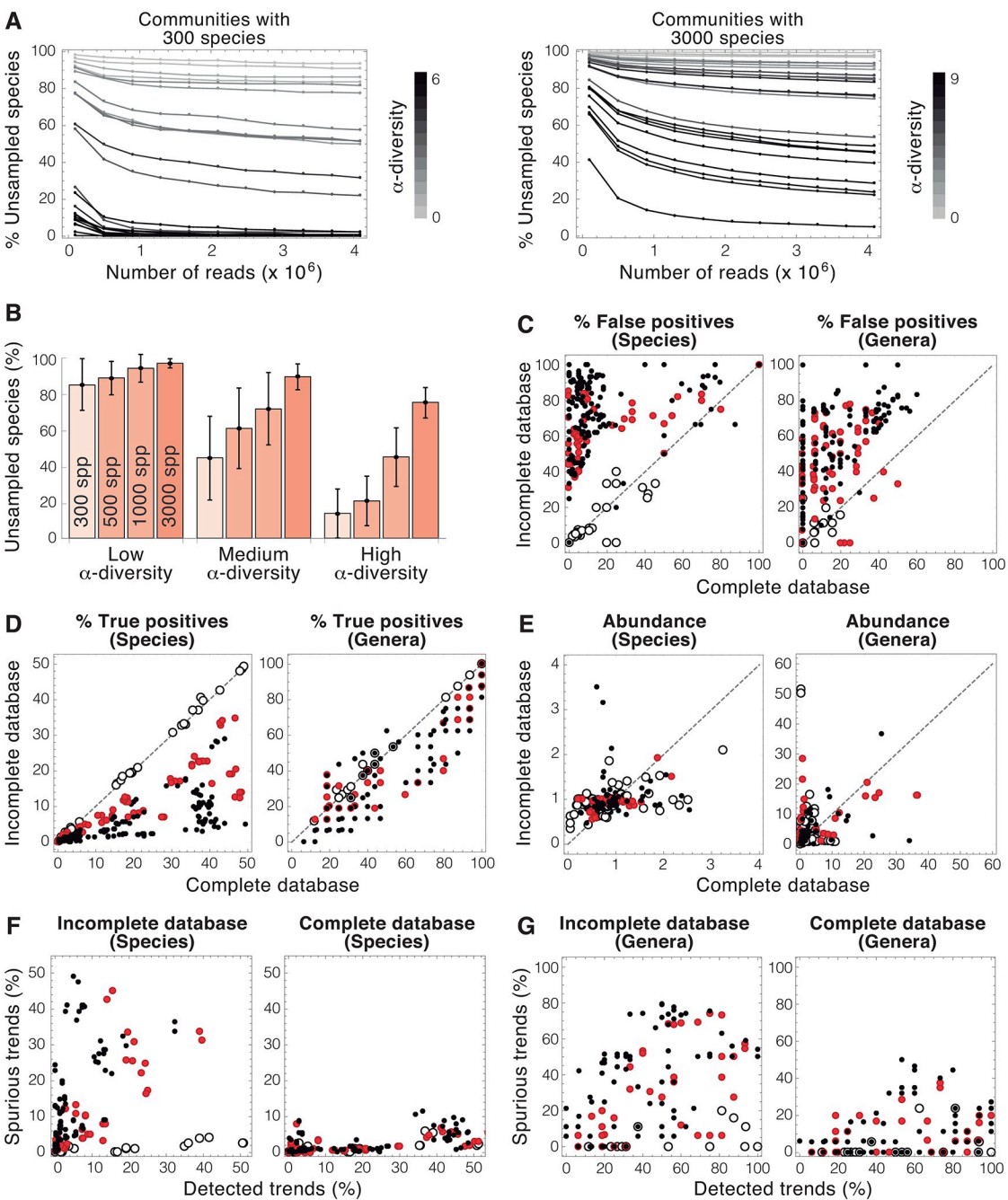

**Fig 2. Evaluation of virtual microbiome characterization by microbiome analyses.** A-B) Effect of ecological parameters (richness of species and $\alpha$-diversity) in the percentage of unsampled species in microbiome analyses ($\alpha$-diversity was calculated using the Shannon index). C-D) Effect of the incompleteness of microbiome databases on the fraction of false positives (D) and true positives (E) at the species (left) and genus (right) level. E) Effect of the incompleteness of databases on the difference between detected and actual abundances of species (left) and genera (right). This difference is defined as $D = 1/N \sum_{i=1}^{N} (\tilde{a}_i - a_i)/a_i$, where $N$ is the number of detected species or genera, and $a_i$ and $\tilde{a}_i$ are the actual and estimated abundances of species or genus $i$ respectively. F-G) Ability of microbiome analysis to detect changes in the abundance of species (F) and genera (G) when incomplete (left) or complete (right) databases are used in the analysis. On the abscissa axis, the percentage of actual changes in abundance in the virtual microbiome detected by the analysis. On the ordinate, the fraction of the changes in abundance found by the analysis that does not correspond to actual changes in the virtual microbiome. (Open dots: 100% of the species or genera of the virtual microbiome are in the databases; Red dots: 50% of species or genera in DBs; Black dots: 25% of species or genera in DBs. Dashed lines: $x = y$.).

actual changes in species abundance in the virtual microbiomes (Fig 2F). Using complete databases greatly reduced the detection of false trends. In any case, less than half of the actual changes in abundance taking place in the virtual microbiomes were detected by genomic analyses. The detection of both actual and spurious trends was greater at the genus level (Fig 2F).

## Overlap between 16S and WGS analysis of virtual microbiomes

To study the differences and similitudes between 16S and WGS analyses, we began by comparing their results when applied to the characterization of the same virtual microbiomes. As a general rule, WGS outperformed 16S in the identification of true positives but 16S analyses detected fewer false positives (see S1 Fig in S1 Appendix). Owing to the problems derived from the use of 16S primers discussed above, the fractions of unidentified species and genera were greater for amplicon than for WGS analyses (S1 Fig in S1 Appendix).

The overlap between 16S and WGS at the species level was below 50% in all the studied virtual microbiomes. In the analyses with less species present in the incomplete databases, this overlap did not reach 10% (Fig 3A). The coincidence between the results of 16S and WGS greatly increased with the amount of information available in the genomic databases. In contrast, using more reads in the analysis did not lead to better 16S-WGS fits (Fig 3B). Unexpectedly, most of the species simultaneously detected by 16S and WGS using incomplete databases

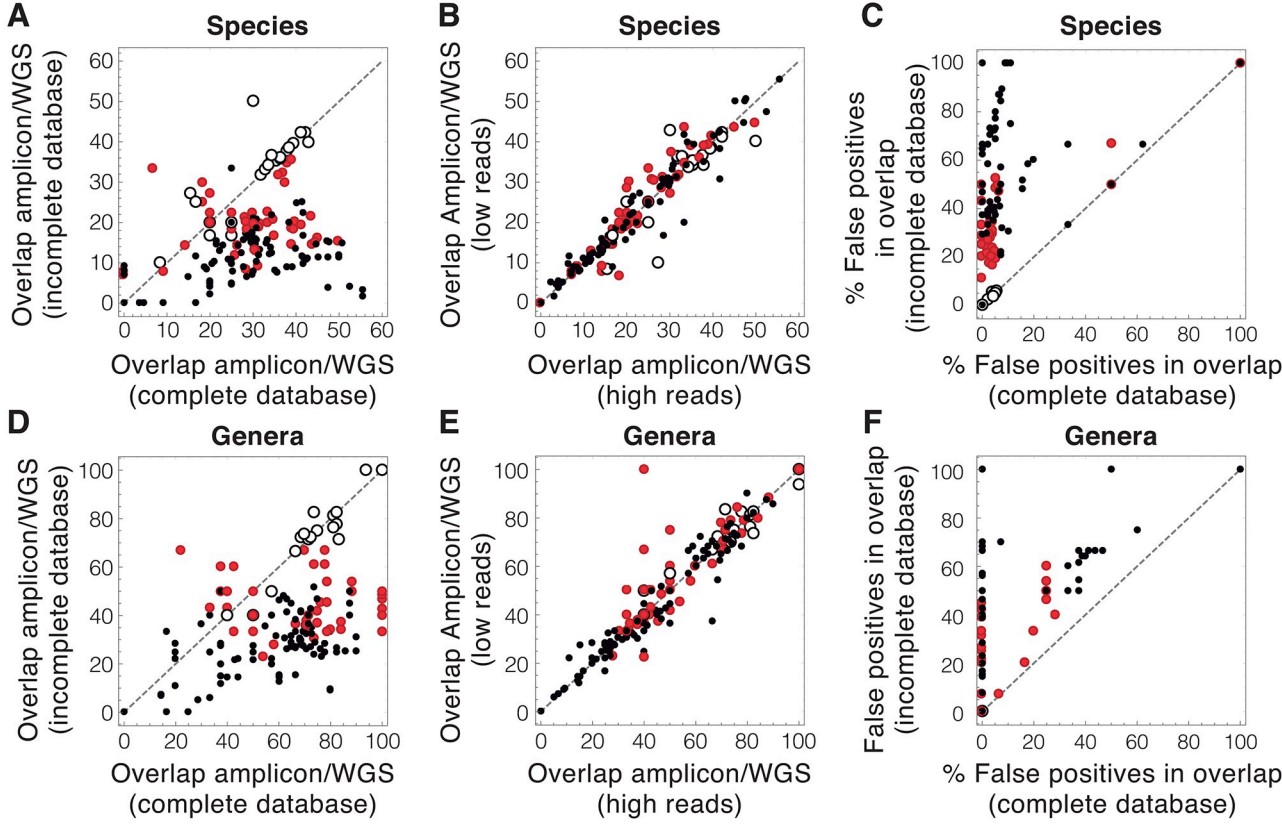

**Fig 3. Overlap between 16S and WGS analyses of virtual microbiomes.** A) Percentage of the species detected by either 16S or WGS that are simultaneously detected by both techniques. B) Effect of the number of reads on the overlap between 16S and WGS. C) Percentage of the species simultaneously detected by 16S and WGS that are not originally present in the virtual microbiome. D-F) Same as A-C for analyses at the genus level. Remark that all the open dots lie at the origin in F. (Open dots: 100% of the species or genera of the virtual microbiome are in the databases; Red dots: 50% of species or genera in DBs; Black dots: 25% of species or genera in DBs. Dashed lines: $x = y$.).

were false positives (Fig 3C). Similar behaviors were observed in the analyses at the genus level (Fig 3D–3F).

These results show that the simultaneous detection of a species or genus by 16S and WGS does not provide additional information about its presence in the virtual microbiome. The list of species and genera found in the intersection between both analyses does not characterize virtual microbiomes more accurately than those found separately by each analysis.

Interestingly, each technique proved capable of identifying species and genera that could not be found by the other (S2A Fig in S1 Appendix). In this regard, the number of true positives detected exclusively by WGS was normally greater than those obtained only by 16S (S2B Fig in S1 Appendix). Still, some species and genera identified by 16S were not detected in WGS analysis. The number of true positives detected only by WGS was greater in analyses with incomplete databases. However, WGS also found more false positives than 16S, an effect that was more pronounced when incomplete databases were used in the analysis (S2B Fig in S1 Appendix).

Altogether, our results point to a very limited ability of amplicon and WGS to accurately characterize virtual microbiomes, a conclusion that can be extrapolated to the study of real-world microbial communities. Since it is obviously impossible to know a priori if a given microbiome is underrepresented in the databanks used in its characterization, there is no way to judge the fraction of false positives found in the analysis, or how many of the taxa present in the microbiome are not identified or even sampled. This entails a high degree of uncertainty in the analysis of microbial communities, an aspect that should be explicitly taken into account in the interpretation of microbiome data. We address this issue in the next section.

### Empirical test of the virtual microbiome predictions in the microbiome of *Galleria mellonella*

To test the predictions of the virtual microbiome models, the microbiome of *Galleria mellonella* larvae fed with a control diet or with polyethylene was investigated using amplicon and WGS. Sixty-eight 16S samples were produced from three tissues, namely Anterior Part (AP), Gut and Silk Glands (SG), control and experimental (11 samples per tissues, plus two more samples for the AP control) producing a total of 22,548,048 MiSeq paired-end 300 bp reads (S1 Data in S2 Appendix). About 50% of the total amount of reads could be classified at Phylum level, with Proteobacteria (52.16% of the reads), Firmicutes (22.97%), and Actinobacteria (15.82%) being the most abundant Phyla, on average (Fig 4A). On the other hand, about 42% of the reads could be classified at genus level with *Pseudomonas* (22.92%), *Enterococcus* (13.70%), *Staphylococcus* (6.75%), *Serratia* (6.73%), and *Acinetobacter* (4.84%) being the most abundant, on average (Fig 4B).

Thirteen samples were also processed with a WGS approach producing a total of 734,523,154 paired-end 150 bp reads (S2 Data in S2 Appendix). After removing the reads that mapped on the *G. mellonella* reference genome, only 221,217 (0.06%) pairs could be classified at Phylum level with Proteobacteria (22.84%), Arthropoda (17.05%), Ascomycota (12.63%), Basidiomycota (9.33%), and Firmicutes (8.19%) being the most abundant, on average (Fig 4A). The presence of Arthropoda suggests that some contamination from the host was still present. At the genus level, a total of 146,444 (0.03%) pairs could be classified, with the most abundant being *Heterobasidion* (Basidiomycota, 12.64%), *Afipia* (Proteobacteria, 12.38%), *Treponema* (Spirochaetes, 10.25%), *Parastagonospora* (Ascomycota, 10.20%), and *Enterococcus* (Firmicutes, 6.53%), on average (Fig 4B).

As a next step, we compared the results of both sequencing strategies, taking into account that the data was produced from the same samples. For this comparison, we only considered

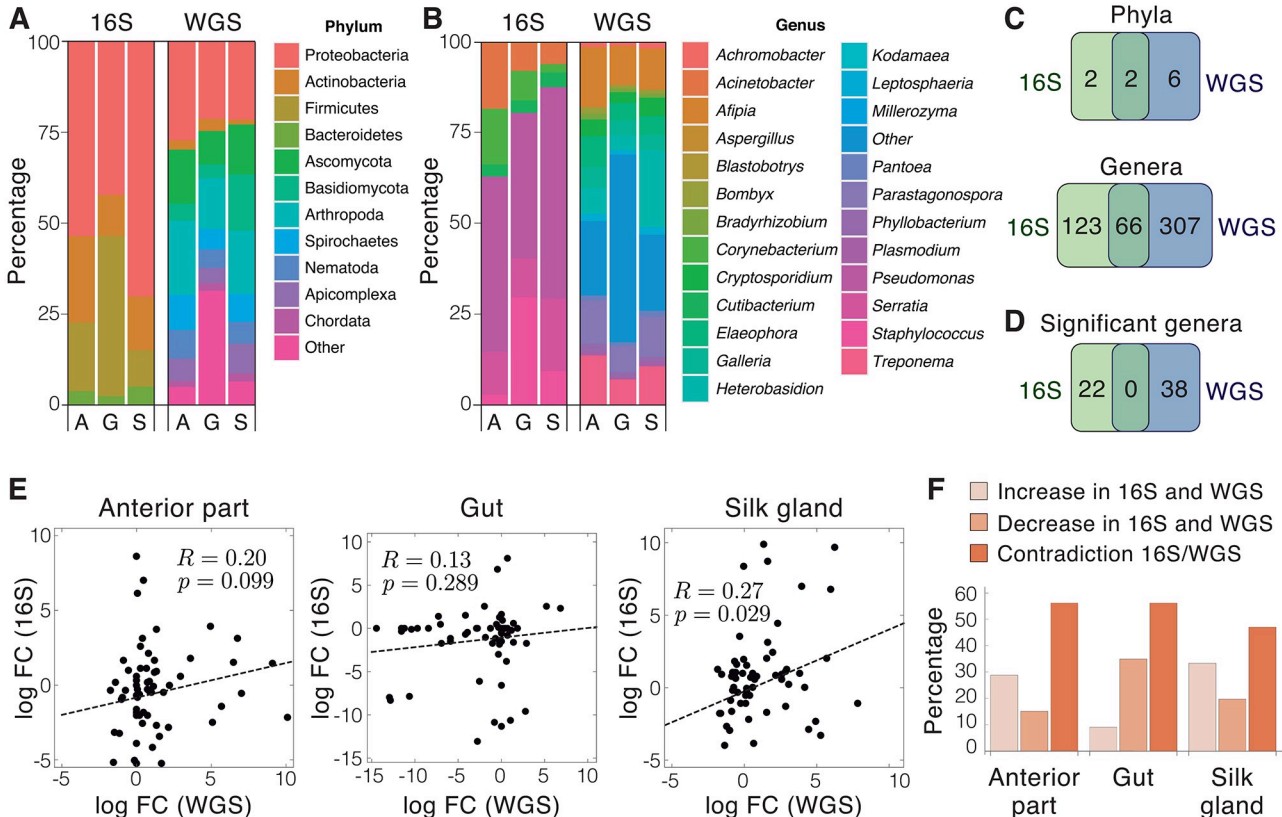

**Fig 4. Overlap between 16S and WGS analyses of the microbiome of *Galleria mellonella* larvae.** A) Barplot showing the percentage of reads classified in the observed Phyla with 16S and WGS respectively in three tissues of *G. mellonella* (A: anterior part; G: gut; S: silk gland). B) Barplot showing the percentage of reads classified in the observed genera with 16S and WGS respectively in three tissues of *G. mellonella* (A: anterior part; G: gut; S: silk gland). C) Venn Diagrams showing the overlap between 16S and WGS. Upper: overlap of the identified phyla; Lower: overlap of the identified genera. D) Overlap of the genera showing significant differences in abundance between treated and control samples across the gut, anterior part, and salivary glands. E) Scatterplots showing the correlation between log₂ fold changes of genera simultaneously detected by WGS and 16S in the anterior part (left), gut (medium), and salivary glands (left) from treated animals with respect to controls. Pearson correlation coefficient (*R*) and its significance (*p*) is reported for each plot. F) Coincidences and discrepancies between changes in abundance detected by 16S and WGS in the anterior part, gut, and silk glands of treated animals with respect to controls.

the taxa with a median abundance higher than 1%. At the phylum level, we observed that only 50% of the phyla detected by 16S could be also identified with the WGS strategy. In addition, the WGS approach detected 6 phyla that were not found by the 16S approach. Considering the total amount of 10 detected phyla, only 2 (20%) could be detected by both methods (Fig 4C). At the genus level, about 35% of the genera detected by 16S sequencing could also be identified by WGS. Considering the total amount of 496 detected genera, only 66 (13.3%) were simultaneously detected by both methods (Fig 4C).

We then performed differential abundance analysis using 16S and WGS data to detect genera whose abundance was affected by the diet. Each tissue was analyzed separately (S1 and S2 Data in S2 Appendix) and gut samples always contained the majority of differential genera. Considering all the tissues, 38 and 22 genera were found to show significant (FDR ≤0.05) differences in abundance with 16S and WGS respectively. Strikingly, none of these differential changes was simultaneously detected by both techniques (Fig 4D).

Even if there was no overlap among significant results, we decided to compare the fold changes detected for common genera between WGS and 16S in order to assess whether the magnitude of change was detected in the same way by the different methods (Fig 4E). Pearson

Correlation of the log2 fold changes in the three tissues ranged from 0.13 to 0.27 between WGS and 16S. Focusing on the trend of the changes (i.e. increase or decrease) without considering their magnitude gave similar results (Fig 4F), with both techniques detecting opposite trends in about half of the cases. The discrepancy between 16S and WGS is even more pronounced for genera showing significant changes in abundance, since the genera found to be significant are not even the same in both cases. Overall, these results point to the fact that we observed a poor agreement in terms of changes in abundance between WGS and amplicon sequencing methods.

## Discussion

In this work, we formulate a computational framework to evaluate the performance of metagenomic analyses based on the generation of virtual microbiomes that simulate real bacterial communities. Using this approach, we identified critical limitations in the capability of currently available technologies to characterize microbiomes. Similar results were obtained using two different tools for taxonomic profiling (S3 Fig in S1 Appendix), which suggests that these limitations may be general and not specific of particular tools. The constrains found within the virtual microbiome framework predicted a poor overlap between WGS and amplicon in the analysis of real-world microbiomes underrepresented in genomic databases. This prediction was confirmed by the discrepancies between both methods in the characterization of the bacteria found in *G. mellonella* larvae.

Some of the limitations of microbiome analysis result from intrinsic features of microbial populations (e.g. the loss of species due to sampling is greater in highly diverse communities). These limitations, imposed by the ecological structure of microbial populations, have been discussed elsewhere [36]. Other constraints arise from the lack of genomic information in the databanks used in metagenomic studies. It is trivial that 16S and WGS analyses cannot detect species that are not represented in genomic databases. Unexpected, however, is that the incompleteness of the available information also leads to an increase in the detection of false positives, which may severely distort the identification of the species present in a microbiome.

It seems natural to assume that applying 16S and WGS methods to characterize the same microbial community should increase the accuracy of the analysis. Intuitively, the detection of the same species by both techniques could be interpreted as the confirmation of its presence in the microbiome. However, our results suggest that this assumption is misleading, since the observed overlap between the results of 16S and WGS can consist mostly of false positives. This result agrees with the poor overlap between 16S and WGS observed in previous studies [33, 34, 50], which, accordingly, could be explained as emerging from the combined effect of the limitations derived from the ecological structure of the microbiome and the insufficiency of the genomic information used in the analyses.

As could be expected, better results can be obtained working at the genus levels, both in terms of microbiome characterization and overlap between 16S and WGS. However, the accuracy of the analyses is still highly dependent on the amount of information present in the databases. In this regard, it is worth noting that the amount of gathered genomic information has grown exponentially in the last decades and that the number of sequence records available at the NCBI databank ranges in the billions (Fig 5A). In contrast, the size of the databases used by metagenomic tools is much smaller owing to obvious computational constrains that limit the volume of information that can be effectively handled in the analyses. This implies that many species or genera present in a given sample will likely be missed by metagenomic analyses, even if those species and genera have been previously described in the literature. The subsequent bias in the analysis critically depends on the metagenomic tool chosen to perform the

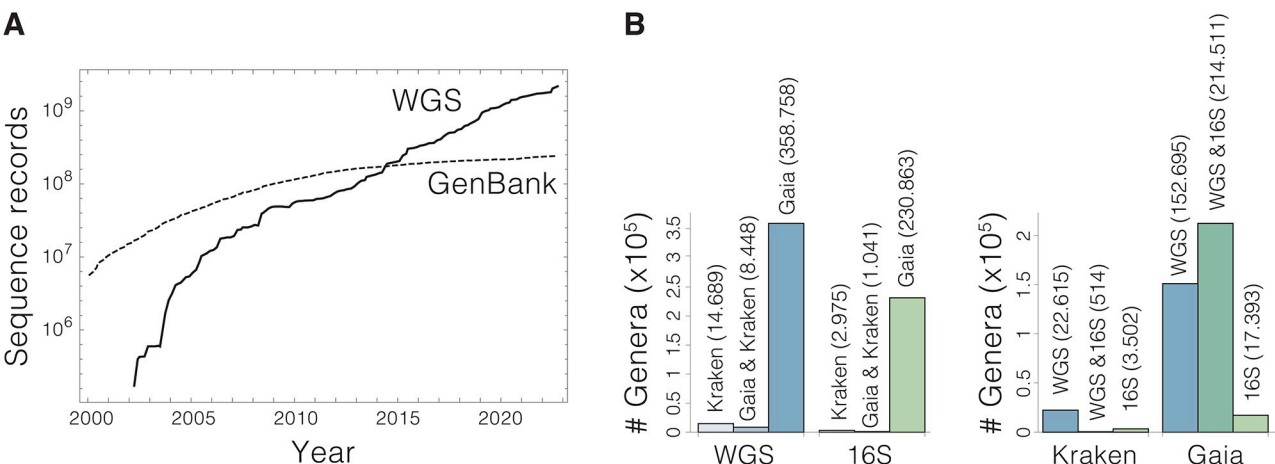

**Fig 5. Information available for metagenomic analyses.** A) The publication of new genomic information in the NCBI databank continues has grown exponentially (data obtained from https://www.ncbi.nlm.nih.gov/genbank/statistics/). B) Metagenomic tools greatly differ not only in the size of their reference databanks but also in the particular genera that are represented in their reference databases. The overlap between the databases used by GAIA and Kraken2 is small in both WGS and 16S (left). In the case of Kraken2, the datasets used in 16S and WGS analyses only coincide in a small fraction of genera (right).

analysis (Fig 5B). Therefore, scenarios in which the 100% of the sample is in represented in the databanks can be rare, specially for less studied microbiomes such as those of insects.

All those limitations critically constrain the successful identification of the bacteria present in different communities. Importantly, they also hinder the detection of the changes in microbial communities induced by external or internal factors. Our results reveal that many of the trends identified at the species level can be spurious. Again, trend detection improves when working at the genus level, but a significant gap still exists between observed and real changes in abundance. This calls into question the approach of studies claiming to find bacteria that proliferate or decline after certain treatments [18, 20, 23, 26–28]. We confirmed this point by studying the microbiome of plastic-fed and control *G. mellonella* larvae. Strikingly, the bacterial genera showing significant changes in abundance in hosts under a diet of plastic were totally different when observed by WGS or amplicon methods. This result questions the widespread experimental strategy of inducing changes in bacterial abundance to detect microorganisms with specific metabolic potentials.

Our results also confirm the utility of the virtual microbiome as a powerful theoretical framework to scrutinize the performance of different techniques of microbiome analysis. The shortcomings of genomic analysis identified within this framework can be addressed in several ways: 1) by working at higher taxonomic levels whenever this is possible. This solution may be insufficient though if further taxonomic precision is needed; 2) continue to accumulate information in genomic databases. This is a fundamental aspect, as the lack of data leads to worse results in terms of characterization and detection of changes in abundance; 3) one must be cautious when interpreting data from metagenomic studies. As has been shown, owing to the limitations inherent to these techniques, their results should not be taken as totally reliable or conclusive.

The ever-growing interest in microbiomes and their potential applications is very much dependent the reliability, richness, and completeness of the databanks available for their accurate description. If microbiome data are to be useful in an effective and reproducible manner, the effort in the field must be channeled towards significantly increasing the amount of available genomic information and finding efficient ways to use this information. We believe

that the virtual microbiome framework can be useful in the design of new approaches to meta-genomic analysis.

## Supporting information

**S1 Appendix.**
(PDF)

**S2 Appendix. List of supplementary data files. S1 Data**: Complete and incomplete databases used in the analyses of virtual microbiomes with GAIA. **S2–S4 Data**: Virtual microbiomes analyzed with GAIA. **S5 Data**: Raw data of the amplicon analysis of *G. mellonella* microbiome with GAIA. **S6 Data**: Raw data of the WGS analysis of *G. mellonella* microbiome with GAIA. **S7 Data**: Significant changes of abundance in *G. mellonella* microbiome detected by 16S analysis with GAIA. **S8 Data**: Significant changes of abundance in *G. mellonella* microbiome detected by WGS analysis with GAIA. **S9 Data**: Virtual microbiomes analyzed with Kraken2. **S10 Data**: Raw data of the analysis of *G. mellonella* gut microbiome with Kraken2.
(ZIP)

## Author Contributions

**Conceptualization:** Riccardo Aiese Cigliano, Clemente F. Arias, Federica Bertocchini.

**Data curation:** Francisco Rodríguez-Ventura, Pere Colomer-Vidal, Federica Bertocchini.

**Formal analysis:** Belén Serrano-Antón, Riccardo Aiese Cigliano, Clemente F. Arias.

**Funding acquisition:** Federica Bertocchini.

**Investigation:** Belén Serrano-Antón, Francisco Rodríguez-Ventura, Pere Colomer-Vidal, Riccardo Aiese Cigliano, Clemente F. Arias, Federica Bertocchini.

**Methodology:** Belén Serrano-Antón.

**Software:** Belén Serrano-Antón, Riccardo Aiese Cigliano.

**Supervision:** Riccardo Aiese Cigliano, Clemente F. Arias, Federica Bertocchini.

**Writing – original draft:** Riccardo Aiese Cigliano, Clemente F. Arias, Federica Bertocchini.

**Writing – review & editing:** Belén Serrano-Antón, Riccardo Aiese Cigliano, Clemente F. Arias, Federica Bertocchini.

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
