## [Decision Letter · Decision Letter 0]

8 Oct 2022

PONE-D-22-19792The virtualome: a computational framework to evaluate microbiome analysesPLOS ONE

Dear Dr. Arias,

Thank you for submitting your manuscript to PLOS ONE. After careful consideration, we feel that it has merit but does not fully meet PLOS ONE’s publication criteria as it currently stands. Therefore, we invite you to submit a revised version of the manuscript that addresses the points raised during the review process.

We look forward to receiving your revised manuscript.

Kind regards,

Patrizia Falabella

Academic Editor

PLOS ONE

Journal Requirements:

2. Please update your submission to use the PLOS LaTeX template. The template and more information on our requirements for LaTeX submissions can be found at http://journals.plos.org/plosone/s/latex

4. Please amend either the abstract on the online submission form (via Edit Submission) or the abstract in the manuscript so that they are identical

"FB and CFA gratefully acknowledge support by the Roechling foundation. BS was partially supported by MINECO grant MTM2017-85020-P."

"FB and CFA gratefully acknowledge support by the Roechling foundation.

BS was partially supported by MINECO grant MTM2017-85020-P."

Reviewers' comments:

Reviewer's Responses to Questions

**Comments to the Author**

1. Is the manuscript technically sound, and do the data support the conclusions?

Reviewer #1: Partly

Reviewer #2: Yes

2. Has the statistical analysis been performed appropriately and rigorously? 

Reviewer #1: N/A

Reviewer #2: Yes

3. Have the authors made all data underlying the findings in their manuscript fully available?

Reviewer #1: Yes

Reviewer #2: Yes

4. Is the manuscript presented in an intelligible fashion and written in standard English?

Reviewer #1: Yes

Reviewer #2: Yes

5. Review Comments to the Author

Reviewer #1: In this work, the authors use a statistical framework to generate virtual microbiomes that serve as benchmarks to compare the performance of amplicon and WGS-based taxonomic profiles. They find that the performance of taxonomic profiling is shckingly poor, especially if the reference datbases are incomplete, but also with complete databases. Moreover, the overlap observed between amplicon and WGS-based profiling is often due to false positives.

The implications of this study, if true, are devastating. What I find most concerning is the poor performance of taxonomic profiling even if the reference database contains all the genomes from the microbiome. Given the strong implications of this result, I find it essential to verify that such poor performance is not just a limitation of the particular tool (GAIA) and parameters used for taxonomic profiling. To that purpose, it would possibly be enough to study what happens when using other popular tools for taxonomic profiling and the complete database (which is the best-case scenario). I understand that the authors may choose not to go that longer way, but then the article should explicitly state that its conclussions are valid for GAIA, but maybe not for other taxonomic profiling algorithms.

From a technical perspective, the statistical properties and algorithms used to generate the virtual microbiome should be described in much greater detail. Given that generation of virtual microbiomes is central to (and possibly the main novelty of) the study, these details could even be presented in the Results section. If the authors prefer to focus the Results section on the comparison of taxonomic profiles, full details should still be provided in the Methods section.

I am not fully aware of Plos ONE policies about code availability; however, following the FAIR principles for open and reproducible research, it would be good to share the code to generate virtual microbiomes in a public repository, such as github or zenodo. Personally, I think that the generation of virtual microbiomes that are compatible with the abundance distributions found in real-world microbiomes is the main scientific breakthrough of this paper and not making the method available substantially diminishes its interest. I appreciate that pseudo-code for the algorithm is included in the Methods, but it needs more detail (what does function "generateAbundances" do? what is "params"?).

Related to the previous issues:

- Discussion of Figures 2C-D require further clarification. Why are there unidentified species when using the complete databases? Do the figures include unsampled species? Are some species being classified as "unidentified" because their assignations are ambiguous? How is it possible that the incomplete database sometimes prudices fewer unidentified genera than the complete database? How is it possible that the complete database produces false positives (perhaps due to an inappropriate management of ambiguous hits)?

- The fact that complete databases often lead to a large fraction of unidentified species and false positives suggests that the bioinformatic workflows used to simulate reads and assign taxonomy are intrinsically inaccurate or not properly optimized for the analysis of virtual microbiomes. Has this poor behavior been previously described for real or mock microbiomes (e.g., in the origial articles of Grinder or GAIA)? If not, how do you explain such poor behavior with virtual microbiomes? The fact that taxonomic profiles are fairly poor even when using the complete databse makes me wonder whether the results with incomplete databases would look better if a software different than GAIA had been used. Do you have any evidence that the findings reported here are qualitatively general and also valid for other taxonomy-assignment workflows?

Minor issues and suggestions:

- The portmanteau "virtualome" seems unncessary and not very intuitive. "Virtual microbiome", though longer, would be more informative and less cumbersome.

- In Section 2.2, first paragraph, sentences following cite [36]: These sentences are somewhat confusing and counterintuitive. The problem is that they refer to two measures of diversity that need to be clearly disambiguated. Species richness (discussed in terms of "small" or "large" communities) and alpha-diversity (discussed just as "diversity") are both measures of diversity, but they affect the percentage of unsampled species in opposite ways: the smaller the species richness, the smaller the unsampled fraction, but the smaller the alpha-diversity, the greater the unsampled fraction. I think that the results would be much more intuitive if ambiguous references to diversity were replaced by more informative terms. Specifically the concept of species richness is broadly known and I see no reason to not use it (instead of talking about small/large communities). As for alpha-diversity, trends would be easier to understand if it was discussed in terms of species (un)evenness, because it is quite intuitive that higher unevenness implies more rare species that may be missed by the analysis.

- Figure 2A: add quantitative legend (color bar) for the shades of gray that correspond to alpha-diversity.

- Page 3: "the simultaneous detection of a species by both techniques does not ensure its positive identification, since the overlap between amplicon and WGS can be dominated by false positives". This requires a citation. If it is a new finding of this study, it should be specified.

Reviewer #2: The manuscript of Serrano-Antón and collaborators entitled "The virtualome: a computational framework to evaluate microbiome analyses" reports the effect of database completeness on the metagenomics analysis outcomes, underlying the critical issues of the current bioinformatics approaches. The manuscript is well-presented, technically precise, and scientifically sound, with the experimental methodology used appropriated for the intended objectives. The presented framework remarked the discrepancies between the Target and the Whole Genome Sequencing approaches both in simulated and real data, taking into account the ecological structure of microbial communities. This study highlights the actual limitations as well as the known advantages of metagenomics investigations, underlying the need for an increased amount of genetics/genomics information in public databases to raise the overall accuracy of real-world data.

I have a few questions for the authors:

Page 5: Authors discussed the "ecological structure of microbial communities imposes a major intrinsic limitation to the capacity of microbiome analyses", but this part is not clear to me. Since the composition of the microbial community can widely vary in real-world data, often without a priori knowledge of the microbial composition under study, the small virtualomes approach tends to reduce the loss of species for technical purposes. In my opinion, the authors should highlight the different findings obtained by the use of the different alpha-diversity databases, as in the discussion the presented differences are not detailed as needed.

Page 6: Regarding the sentence "Remarkably, the incompleteness of the databanks affected the analyses of virtualomes even when the 100% of their species were represented in the incomplete databases (the open dots do not lie in the diagonal in Fig. 2D)." how did the authors interpret these findings? The authors should add more details on their evaluation of the reported findings.

Page 7: In the sentence "the number of true positives detected exclusively by WGS was normally greater than those obtained only by 16S" the authors highlight the difference between 16S and WGS results. Did the authors find any statistically significant difference between the number of real positives found by the use of Target compared to the use of WGS? What about the false positive?

Page 9: "The discrepancy between 16S and WGS is pronounced for genera", so even by the use of Pearson Correlation, readers cannot properly get which method could be the best in the study of an unknown microbial community. In my opinion, authors should highlight the best and the worst of each technique, even by adding a summary table reporting the main findings in both real and simulated data, to help readers in understanding of the presented results.

Page 13: is not clear to me if all reads that did not map to the G. mellonella reference genome were successfully classified by the use of the GAIA software, or if it was necessary to assemble and classify any Metagenome Assembled Genomes (MAGs). Can the authors clarify this point?

6. PLOS authors have the option to publish the peer review history of their article (what does this mean?). If published, this will include your full peer review and any attached files.

Reviewer #1: No

Reviewer #2: No

---

## [Author Response · Author response to Decision Letter 0]

7 Dec 2022

Please find below a point-by-point response (in blue) to the questions posed by the Reviewers (in black).

Reviewer #1: In this work, the authors use a statistical framework to generate virtual microbiomes that serve as benchmarks to compare the performance of amplicon and WGS-based taxonomic profiles. They find that the performance of taxonomic profiling is shckingly poor, especially if the reference datbases are incomplete, but also with complete databases. Moreover, the overlap observed between amplicon and WGS-based profiling is often due to false positives.  The implications of this study, if true, are devastating. What I find most concerning is the poor performance of taxonomic profiling even if the reference database contains all the genomes from the microbiome. Given the strong implications of this result, I find it essential to verify that such poor performance is not just a limitation of the particular tool (GAIA) and parameters used for taxonomic profiling. To that purpose, it would possibly be enough to study what happens when using other popular tools for taxonomic profiling and the complete database (which is the best-case scenario). I understand that the authors may choose not to go that longer way, but then the article should explicitly state that its conclussions are valid for GAIA, but maybe not for other taxonomic profiling algorithms. 

We have repeated the analysis of the best-case scenario (100% of the species and genera in the database) using Kraken2, which is one of the most used algorithms in the field and one of the few that can process both amplicon and WGS-based metagenomics data as GAIA. To do that, we have generated new virtualomes, analogous to those considered in the manuscript, using species and genera taken from the databases used by Kraken2 (we will return to this point below). The performance of Kraken2 in terms of false positives and poor overlap between 16S and WGS analyses is similar to that of GAIA, which suggests that the limitations identified in our work are not specific of GAIA (these new results are shown in a new supplementary figure, Supp. Fig. 1.A-C). In particular, it is remarkable that an important fraction of the genera simultaneously found in 16S and WGS analyses using Kraken2 are actually false positives.

We have also analyzed the biological samples taken from Galleria with Kraken2. In this case, the overlap between 16S and WGS analyses is around 10% at the genus level, which is also similar to that found with GAIA. We have included this new result in Supp. Fig. 1.D). We have changed the Methods section to add the details of the Kraken analyses.

It is worth noting that many of the species and genera of the original virtualomes are not present in the databanks used by Kraken2 (this is why we created new ad hoc virtualomes for the analysis with Kraken2). This would obviously constrain the ability of Kraken2 to characterize the original virtual communities. In fact, it is difficult to find enough species and genera simultaneously present in databases used by GAIA and Kraken to simulate the 100% scenario with both tools at the same time. This remark led us to take a closer look at the information that is actually used in metagenomics analyses. To our surprise, the databanks used by GAIA and Kraken2 are very different from each other, i.e. the number of species and genera in the intersection between both databanks is small. This implies that it is likely that some species or genera will not be present in the databases used in metagenomic analysis, even if those species and genera have been previously described. The choice of a metagenomic tool entails the loss of species and genera from the reference databank. In fact, the number of sequences contained in the NCBI database is in the range of billions. No metagenomic tool can manage this volume of information to date, which means that the databases used in the analyses only contain a small fraction of the genomes that have been described to date. We have included a new figure in the text and a brief discussion on this point (Figure 5):

"In this regard, it is worth noting that the amount of gathered genomic information has grown exponentially in the last decades and that the number of sequence records available at the NCBI databank ranges in the billions (Fig. 5.A). In contrast, the size of the databases used by metagenomic tools is much smaller owing to obvious computational constrains that limit the volume of information that can be effectively handled in the analyses. This implies that many species or genera present in a given sample will likely be missed by metagenomic analyses, even if those species and genera have been previously described in the literature. The subsequent bias in the analysis critically depends on the metagenomic tool chosen to perform the analysis (Fig. 5.B). Therefore, scenarios in which the 100% of the sample is in represented in the databanks can be rare, specially for less studied microbiomes such as those of insects." (lines 485-496)

We have also added a phrase about this in the Discussion:

"If microbiome data are to be useful in an effective and reproducible manner, the effort in the field must be channeled towards significantly increasing the amount of available genomic information and finding efficient ways to use this information. We believe that the virtualome framework can be useful in the design of new approaches to metagenomic analysis." (pages 525-528)

 From a technical perspective, the statistical properties and algorithms used to generate the virtual microbiome should be described in much greater detail. Given that generation of virtual microbiomes is central to (and possibly the main novelty of) the study, these details could even be presented in the Results section. If the authors prefer to focus the Results section on the comparison of taxonomic profiles, full details should still be provided in the Methods section. 

We have described the algorithms in much more detail in the Method section and in (link). We hope the new presentation will be satisfactory.

 I am not fully aware of Plos ONE policies about code availability; however, following the FAIR principles for open and reproducible research, it would be good to share the code to generate virtual microbiomes in a public repository, such as github or zenodo. Personally, I think that the generation of virtual microbiomes that are compatible with the abundance distributions found in real-world microbiomes is the main scientific breakthrough of this paper and not making the method available substantially diminishes its interest. I appreciate that pseudo-code for the algorithm is included in the Methods, but it needs more detail (what does function "generateAbundances" do? what is "params"?). 

We agree, see the response to the previous point.

 Related to the previous issues:  - Discussion of Figures 2C-D require further clarification. Why are there unidentified species when using the complete databases? Do the figures include unsampled species? Are some species being classified as "unidentified" because their assignations are ambiguous? How is it possible that the incomplete database sometimes prudices fewer unidentified genera than the complete database? How is it possible that the complete database produces false positives (perhaps due to an inappropriate management of ambiguous hits)? 

We thank the reviewer for this remark. In fact, the calculation of unidentified species was not accurate. This was an indirect estimation using the expected abundances, and the procedure to do that was not correct. Unfortunately, we did not save the information necessary to measure this variable accurately, so we have decided to remove this result from the text (originally shown in Figure 2C). In any case, this does not affect the rest of the results and the conclusions of our work.

 - The fact that complete databases often lead to a large fraction of unidentified species and false positives suggests that the bioinformatic workflows used to simulate reads and assign taxonomy are intrinsically inaccurate or not properly optimized for the analysis of virtual microbiomes. Has this poor behavior been previously described for real or mock microbiomes (e.g., in the origial articles of Grinder or GAIA)? If not, how do you explain such poor behavior with virtual microbiomes? The fact that taxonomic profiles are fairly poor even when using the complete databse makes me wonder whether the results with incomplete databases would look better if a software different than GAIA had been used. Do you have any evidence that the findings reported here are qualitatively general and also valid for other taxonomy-assignment workflows? 

As explained above, we have obtained similar results using Kraken2. In the manuscript, we cite some studies that show that analyzing the same samples using different techniques yield contradictory results (poor overlaps between alternative analysis of the same samples), which suggests that doubts about the accuracy of metagenomic analysis have already been raised in the field (see references 33-35 in the manuscript).

 Minor issues and suggestions:  - The portmanteau "virtualome" seems unnecessary and not very intuitive. "Virtual microbiome", though longer, would be more informative and less cumbersome.

We do not agree with the Reviewer in this point. In our opinion, the term "virtualome" is intuitive and less cumbersome than "virtual microbiome".

 - In Section 2.2, first paragraph, sentences following cite [36]: These sentences are somewhat confusing and counterintuitive. The problem is that they refer to two measures of diversity that need to be clearly disambiguated. Species richness (discussed in terms of "small" or "large" communities) and alpha-diversity (discussed just as "diversity") are both measures of diversity, but they affect the percentage of unsampled species in opposite ways: the smaller the species richness, the smaller the unsampled fraction, but the smaller the alpha-diversity, the greater the unsampled fraction. I think that the results would be much more intuitive if ambiguous references to diversity were replaced by more informative terms. Specifically the concept of species richness is broadly known and I see no reason to not use it (instead of talking about small/large communities). As for alpha-diversity, trends would be easier to understand if it was discussed in terms of species (un)evenness, because it is quite intuitive that higher unevenness implies more rare species that may be missed by the analysis. 

We agree with the Reviewer that this paragraph was confusing. We have reformulated it to:

"To test the effect of sampling on the performance of microbiome analysis, we generated communities that differ in both species richness and α-diversity. Smaller species richness and greater α-diversities resulted in smaller fractions of unsampled species (Fig. 2A left). In contrast, low-diversity communities are dominated by a few species or genera. These unevenness in the relative abundance implies that rare species are likely to be missed by the analysis. In these cases, increasing the number of reads only led to a moderate growth in the percentage of species detected (light grey lines, corresponding to this type of communities, do not fall below 80% in 2A). This effect was more pronounced in more species-rich populations, in which greater sampling efforts did not necessarily result in a substantial reduction in the number of unsampled species, even for high values of α-diversity (Fig. 2A right). The ecological structure of microbial communities (in terms of richness of species and α-diversity) imposes a major intrinsic limitation to the capacity of microbiome analyses, as shown by the loss of over 80% of the species in virtualomes with low diversity (Fig.2B). In the subsequent evaluation of microbiome analyses, we used communities with low species richness (containing only 300 species) to minimize the fraction of unsampled species". (lines 309-324)

 - Figure 2A: add quantitative legend (color bar) for the shades of gray that correspond to alpha-diversity. 

Done.

 - Page 3: "the simultaneous detection of a species by both techniques does not ensure its positive identification, since the overlap between amplicon and WGS can be dominated by false positives". This requires a citation. If it is a new finding of this study, it should be specified.

We have reformulated this to make it explicit that this is a result of this study:

"Using this approach, we have identified the incompleteness of databases as a key constraint to the accuracy of microbiome analyses. Our results suggest that the poor overlap between amplicon and WGS already encountered with human samples does not seem to be an exception. Furthermore, we show in this work that the simultaneous detection of a species by both techniques does not ensure its positive identification, since the overlap between amplicon and WGS can be dominated by false positives, i.e. by species that are not present in the original samples". (lines 97-102)

\fReviewer #2:

 The manuscript of Serrano-Antón and collaborators entitled "The virtualome: a computational framework to evaluate microbiome analyses" reports the effect of database completeness on the metagenomics analysis outcomes, underlying the critical issues of the current bioinformatics approaches. The manuscript is well-presented, technically precise, and scientifically sound, with the experimental methodology used appropriated for the intended objectives. The presented framework remarked the discrepancies between the Target and the Whole Genome Sequencing approaches both in simulated and real data, taking into account the ecological structure of microbial communities. This study highlights the actual limitations as well as the known advantages of metagenomics investigations, underlying the need for an increased amount of genetics/genomics information in public databases to raise the overall accuracy of real-world data. 

We thank the Reviewer for his/her positive comments.

 I have a few questions for the authors:  Page 5: Authors discussed the "ecological structure of microbial communities imposes a major intrinsic limitation to the capacity of microbiome analyses", but this part is not clear to me. Since the composition of the microbial community can widely vary in real-world data, often without a priori knowledge of the microbial composition under study, the small virtualomes approach tends to reduce the loss of species for technical purposes. In my opinion, the authors should highlight the different findings obtained by the use of the different alpha-diversity databases, as in the discussion the presented differences are not detailed as needed. 

That the ecological structure of microbial communities can affect metagenomic analyses is not a new result (see for instance reference 36 in the text). In our analyses, the choice of virtualomes with low species richness is not motivated by technical reasons but by the relative lower fraction of unsampled species and genera in this type of communities (as shown in Fig. 2A-B). This choice is intended as defining a lower boundary to the potential inaccuracies of the subsequent metagenomic analyses of the virtualomes. For instance, since the fraction of unsampled species and genera will be greater in more species-rich communities, the fraction of true positives will be smaller. 

We do not understand the question of the alpha-diversity of the databases. This ecological feature depends on the relative abundance of species and genera in a community. In the databases, each species and genus can only be present or absent, so there is no space for relative abundance. The databases used in this work were created to contain different fractions of species and genera present in the virtualomes, and the effect of this factor is discussed in depth in the section entitled "Effect of the incompleteness of databases in the characterization of virtualomes by microbiome analyses", 

 Page 6: Regarding the sentence "Remarkably, the incompleteness of the databanks affected the analyses of virtualomes even when the 100% of their species were represented in the incomplete databases (the open dots do not lie in the diagonal in Fig. 2D)." how did the authors interpret these findings? The authors should add more details on their evaluation of the reported findings.

We have included a more detailed interpretation of this result in the text:

"The detection of false positives is determined by the presence in the reference databank of similar sequences to those sampled from the virtualomes. Greater databanks may contain more sequences exhibiting similarities with virtualome sequences, thus increasing the chances of misidentification. Conversely, if the number of potential candidates decreases, so can do the number of false positives." (lines 335-339)

 Page 7: In the sentence "the number of true positives detected exclusively by WGS was normally greater than those obtained only by 16S" the authors highlight the difference between 16S and WGS results. Did the authors find any statistically significant difference between the number of real positives found by the use of Target compared to the use of WGS? What about the false positive?

We did not perform any statistical analysis to compare the results of alternative virtualome analysis. The differences are sufficiently clear and the use of statistics would not add any valuable information. On the other hand, we did not intend to compare the efficiency of 16S vs WGS approaches, only to highlight potential limitations of metagenomic analyses in general.

 Page 9: "The discrepancy between 16S and WGS is pronounced for genera", so even by the use of Pearson Correlation, readers cannot properly get which method could be the best in the study of an unknown microbial community. In my opinion, authors should highlight the best and the worst of each technique, even by adding a summary table reporting the main findings in both real and simulated data, to help readers in understanding of the presented results. 

As noted in the previous point, we did not intend to compare different metagenomic techniques or tools. In fact, there is probably no single criterium to decide what "better" means in this context. Should we try to reduce the number of false positives? Increase the fraction of true positives? Reduce the cost of the analysis? In fact, we believe that the virtualome provides a useful framework in which these and other interesting issues could be addressed but those specific analyses are beyond the scope of this work. The optimal performance of metagenomic analyses is taken for granted in the field. The virtualomes show that these analyses should be interpreted with care.

 Page 13: is not clear to me if all reads that did not map to the G. mellonella reference genome were successfully classified by the use of the GAIA software, or if it was necessary to assemble and classify any Metagenome Assembled Genomes (MAGs). Can the authors clarify this point?

Only a small fraction of the reads that did not map to the G. mellonella reference genome could be classified. As indicated in the text (line 417), on average only 0.06% of the filtered reads could be classified with GAIA. We obtained similar results with Kraken2 (results not showed). We attempted to generated MAGs from the reads however we did not obtain high quality genomes and most of the contigs remained unclassified as well.

---

## [Decision Letter · Decision Letter 1]

22 Dec 2022

PONE-D-22-19792R1The virtualome: a computational framework to evaluate microbiome analysesPLOS ONE

Dear Dr. Arias,

Thank you for submitting your manuscript to PLOS ONE. After careful consideration, we feel that it has merit but does not fully meet PLOS ONE’s publication criteria as it currently stands. Therefore, we invite you to submit a revised version of the manuscript that addresses the points raised during the review process.

We look forward to receiving your revised manuscript.

Kind regards,

Patrizia Falabella

Academic Editor

PLOS ONE

Journal Requirements:

Reviewers' comments:

Reviewer's Responses to Questions

**Comments to the Author**

1. If the authors have adequately addressed your comments raised in a previous round of review and you feel that this manuscript is now acceptable for publication, you may indicate that here to bypass the “Comments to the Author” section, enter your conflict of interest statement in the “Confidential to Editor” section, and submit your "Accept" recommendation.

Reviewer #1: (No Response)

Reviewer #2: All comments have been addressed

2. Is the manuscript technically sound, and do the data support the conclusions?

Reviewer #1: Yes

Reviewer #2: Yes

3. Has the statistical analysis been performed appropriately and rigorously? 

Reviewer #1: N/A

Reviewer #2: Yes

4. Have the authors made all data underlying the findings in their manuscript fully available?

Reviewer #1: No

Reviewer #2: Yes

5. Is the manuscript presented in an intelligible fashion and written in standard English?

Reviewer #1: No

Reviewer #2: Yes

6. Review Comments to the Author

Reviewer #1: In the revised version of the manuscript, the authors made a remarkable effort to show the generality of their results by reproducing some parts of the analysis with a different taxonomic profiling tool (Kraken). Some obscure claims regarding unidentified taxa have been removed and the origin of false positives is now clearly explained. Overall, these changes notably improved the quality of the manuscript and the robustness of its conclusions.

I appreciate the authors' effort to better describe the algorithm that produces virtual microbiomes. I still do not understand why they did not release a public, functional implementation of their code, although I leave it to the Editor to decide whether that is a requisite for publication. Anyway, the pseudocode included in the methods section is still incomplete: there are 4 subroutines in EcologyGeneration that are not proprely described. Furthermore, I could not find the parameter values (mu, sigma, shape, scale) used to generate the abundance matrices.

I understand that, from the authors' perspective, it is hard to let go of a fancy word like "virtualome", which is certainly catchy. However, I must insist that it is nonsense. There are already many "omes" in the recent biological literature, perhaps too many, but at least most of them are self-explanatory and semantically coherent. Genomes, proteomes, transcriptomes, interactomes, and microbiomes (to name a few) clearly refer to sets of genes, proteins, transcripts, interactions, and microbes, respectively. Accordingly, virtualomes would be sets of some kind of virtual entities. What does that mean? I could think of something like a set of virtual environments in a computing context, although that would be quite a stretch. However, the authors use the term to refer to a virtual microbiome (I guess that the rationale is to combine VIRTUAL with microbiOME). From a semantic perspective, that is nonsense. Following that rule, it could also mean virtual proteome, virtual interactome, virtual transcriptome, virtual metabolome... Or none of that. It is perhaps a funny word to describe the tool in an informal setting, among colleagues or perhaps as a joke in a conference if you want to get the attention of the audience. But I hardly see how it would be an acceptable (that is, unambiguous and semantically coherent) scientific term.

Some typos:

239: "does to distinguish" -> "does NOT distinguish"

325: "These unevenness in the relative abundance implies". Remove "These"

Reviewer #2: In this last version, the authors addressed all the concerns raised by the reviewers, extending certain sections and clarifying important aspects of the actual limitations of the metagenomics analysis. In particular, the new analysis performed using the Kraken2 well supported the results and conclusions of the manuscripts.

7. PLOS authors have the option to publish the peer review history of their article (what does this mean?). If published, this will include your full peer review and any attached files.

Reviewer #1: No

Reviewer #2: No

---

## [Author Response · Author response to Decision Letter 1]

27 Dec 2022

Please find below a point-by-point response (in blue) to the questions posed by the Reviewers (in black).

Reviewer #1: In the revised version of the manuscript, the authors made a remarkable effort to show the generality of their results by reproducing some parts of the analysis with a different taxonomic profiling tool (Kraken). Some obscure claims regarding unidentified taxa have been removed and the origin of false positives is now clearly explained. Overall, these changes notably improved the quality of the manuscript and the robustness of its conclusions. 

We thank the Reviewer for his/her positive comments and for his/her contribution to the improvement of this article.

 I appreciate the authors' effort to better describe the algorithm that produces virtual microbiomes. I still do not understand why they did not release a public, functional implementation of their code, although I leave it to the Editor to decide whether that is a requisite for publication. Anyway, the pseudocode included in the methods section is still incomplete: there are 4 subroutines in EcologyGeneration that are not proprely described. Furthermore, I could not find the parameter values (mu, sigma, shape, scale) used to generate the abundance matrices.

We forgot to add the link to the public repository where the code is available (https://github.com/bserranoanton/Virtual-microbiome). I apologize for this error. We have included it in the revised version of the text. In this site it is also possible to access the values of the parameters used to generate each virtual microbiome. 

"The full implementation of the algorithm can be found at the following link https://github.com/bserranoanton/Virtual-microbiome. In addition, information on all the parameters used for the generation of the abundance matrices of the virtual microbiomes can also be accessed." (lines 139-142).

 I understand that, from the authors' perspective, it is hard to let go of a fancy word like "virtualome", which is certainly catchy. However, I must insist that it is nonsense. There are already many "omes" in the recent biological literature, perhaps too many, but at least most of them are self-explanatory and semantically coherent. Genomes, proteomes, transcriptomes, interactomes, and microbiomes (to name a few) clearly refer to sets of genes, proteins, transcripts, interactions, and microbes, respectively. Accordingly, virtualomes would be sets of some kind of virtual entities. What does that mean? I could think of something like a set of virtual environments in a computing context, although that would be quite a stretch. However, the authors use the term to refer to a virtual microbiome (I guess that the rationale is to combine VIRTUAL with microbiOME). From a semantic perspective, that is nonsense. Following that rule, it could also mean virtual proteome, virtual interactome, virtual transcriptome, virtual metabolome... Or none of that. It is perhaps a funny word to describe the tool in an informal setting, among colleagues or perhaps as a joke in a conference if you want to get the attention of the audience. But I hardly see how it would be an acceptable (that is, unambiguous and semantically coherent) scientific term.

The Reviewer has provided solid arguments against the use of the word "virtualome". We have changed it accordingly throughout the text, including the title, and in the corresponding figures.  Some typos: 239: "does to distinguish" -> "does NOT distinguish" 325: "These unevenness in the relative abundance implies". Remove "These"

Corrected.

Reviewer #2: In this last version, the authors addressed all the concerns raised by the reviewers, extending certain sections and clarifying important aspects of the actual limitations of the metagenomics analysis. In particular, the new analysis performed using the Kraken2 well supported the results and conclusions of the manuscripts.

We thank the Reviewer for his/her positive evaluation of this work.

---

## [Decision Letter · Decision Letter 2]

28 Dec 2022

The virtual microbiome: a computational framework to evaluate microbiome analyses

PONE-D-22-19792R2

Dear Dr. Arias,

We’re pleased to inform you that your manuscript has been judged scientifically suitable for publication and will be formally accepted for publication once it meets all outstanding technical requirements.

Kind regards,

Patrizia Falabella

Academic Editor

PLOS ONE

Additional Editor Comments (optional):

Reviewers' comments:

Reviewer's Responses to Questions

**Comments to the Author**

1. If the authors have adequately addressed your comments raised in a previous round of review and you feel that this manuscript is now acceptable for publication, you may indicate that here to bypass the “Comments to the Author” section, enter your conflict of interest statement in the “Confidential to Editor” section, and submit your "Accept" recommendation.

Reviewer #1: All comments have been addressed

2. Is the manuscript technically sound, and do the data support the conclusions?

Reviewer #1: Yes

3. Has the statistical analysis been performed appropriately and rigorously? 

Reviewer #1: N/A

4. Have the authors made all data underlying the findings in their manuscript fully available?

Reviewer #1: Yes

5. Is the manuscript presented in an intelligible fashion and written in standard English?

Reviewer #1: Yes

6. Review Comments to the Author

Reviewer #1: All my comments have been addressed. Therefore, I support publication of this work in Plos One. I would want to congratulate the authors for this fine piece of research, which will help guide future taxonomic profiling works based on metagenomics.

7. PLOS authors have the option to publish the peer review history of their article (what does this mean?). If published, this will include your full peer review and any attached files.

Reviewer #1: No

---

## [Editor Report · Acceptance letter]

4 Jan 2023

PONE-D-22-19792R2 

The virtual microbiome: a computational framework to evaluate microbiome analyses 

Dear Dr. Arias:

I'm pleased to inform you that your manuscript has been deemed suitable for publication in PLOS ONE. Congratulations! Your manuscript is now with our production department. 

Kind regards, 

on behalf of

Prof. Patrizia Falabella 

Academic Editor

PLOS ONE